# In Vitro Release and In Vivo Pharmacokinetics of Praziquantel Loaded in Different Polymer Particles

**DOI:** 10.3390/ma16093382

**Published:** 2023-04-26

**Authors:** Emiliane Daher Pereira, Luciana da Silva Dutra, Thamiris Franckini Paiva, Larissa Leite de Almeida Carvalho, Helvécio Vinícius Antunes Rocha, José Carlos Pinto

**Affiliations:** 1Programa de Engenharia Química/COPPE, Universidade Federal do Rio de Janeiro, Cidade Universitária, CP: 68502, Rio de Janeiro 21941-972, RJ, Brazil; 2SENAI CETIQT, Instituto SENAI de Inovação em Biossintéticos e Fibras, Cidade Universitária, Rua Fernando de Souza Barros, Rio de Janeiro 21941-857, RJ, Brazil; 3Programa de Engenharia de Processos Químicos e Bioquímicos/EQ, Universidade Federal do Rio de Janeiro, Cidade Universitária, Rio de Janeiro 21949-900, RJ, Brazil; 4Laboratório de Micro e Nanotecnologia, Instituto de Tecnologia de Fármacos—Farmanguinhos, Fundação Oswaldo Cruz, Rio de Janeiro 21040-361, RJ, Brazil

**Keywords:** praziquantel, polymer systems, PMMA, microparticles, nanoparticles, in vitro study, in vivo pharmacokinetics

## Abstract

Approximately 1 billion people are affected by neglected diseases around the world. Among these diseases, schistosomiasis constitutes one of the most important public health problems, being caused by *Schistosoma mansoni* and treated through the oral administration of praziquantel (PZQ). Despite being a common disease in children, the medication is delivered in the form of large, bitter-tasting tablets, which makes it difficult for patients to comply with the treatment. In order to mask the taste of the drug, allow more appropriate doses for children, and enhance the absorption by the body, different polymer matrices based on poly(methyl methacrylate) (PMMA) were developed and used to encapsulate PZQ. Polymer matrices included PMMA nano- and microparticles, PMMA-co-DEAEMA (2-(diethylamino)ethyl methacrylate), and PMMA-co-DMAEMA (2-(dimethylamino)ethyl methacrylate) microparticles. The performances of the drug-loaded particles were characterized in vitro through dissolution tests and in vivo through pharmacokinetic analyses in rats for the first time. The in vitro dissolution studies were carried out in accordance with the Brazilian Pharmacopeia and revealed a good PZQ release profile in an acidic medium for the PMMA-DEAEMA copolymer, reaching values close to 100 % in less than 3 h. The in vivo pharmacokinetic analyses were conducted using free PZQ as the control group that was compared with the investigated matrices. The drug was administered orally at doses of 60 mg/kg, and the PMMA-co-DEAEMA copolymer microparticles were found to be the most efficient release system among the investigated ones, reaching a C_max_ value of 1007 ± 83 ng/mL, even higher than that observed for free PZQ, which displayed a C_max_ value of 432 ± 98 ng/mL.

## 1. Introduction

Neglected diseases affect about 1 billion people in the world, and among these, schistosomiasis constitutes a major public health problem [1]. Data from the World Health Organization (WHO) indicate that at least 251.4 million people needed preventive treatment for schistosomiasis in 2021, although only about 30% of these received treatment. Additionally, according to the WHO, in tropical and subtropical areas, schistosomiasis can be considered the second-most important disease both in socioeconomic and public health terms, surpassed only by malaria [2].

Schistosomiasis is caused by a parasite called *Schistosoma mansoni,* and praziquantel (PZQ) is the drug of choice for this treatment [3,4]. Although schistosomiasis is a common disease in children, this drug product is delivered in the form of large, bitter-tasting tablets, which makes it difficult to comply with the treatment [5,6].

In order to mask the taste of PZQ, allow more appropriate doses for children, and enhance the drug’s absorption by the body, different approaches can be used, including drug complexation, micro- or nanoencapsulation, hot melt extrusion, coating, granulation, mixing with sweeteners, lyophilization, and printing, among others [7,8]. In addition to these traditional methods, new techniques based on the use of supercritical CO_2_ have been reported in the literature as promising alternatives for replacing commonly used toxic organic solvents in release applications. In this scenario, supercritical CO_2_ technologies can be considered cleaner and more sustainable strategies for the encapsulation of several active principles [9,10].

Particularly, polymer encapsulation encompasses a set of techniques that can be used to encapsulate or coat bioactive compounds with a determined physical system [11]. In this case, the release of the encapsulated material is of fundamental importance and can be manipulated to act in specific places of the body for certain specified time intervals and at a determined speed. In particular, drug delivery systems can promote (i) the controlled release of encapsulated materials, protecting them from degradation (caused by moisture, light, heat, and/or oxidation) and ensuring a longer shelf life; (ii) the masking of undesired organoleptic properties of the encapsulated materials, such as color, taste, and odor; (iii) improved dispersion of water-insoluble substances in aqueous media; and (iv) the reduction or elimination of gastric irritation or side effects arising from some drugs [12].

Among the many distinct polymers that can be used for the manufacture of drug delivery systems, poly(methyl methacrylate) (PMMA) and its copolymers have been widely studied because PMMA is a bio-inert material, presents low toxicity, is highly compatible with many drugs, and is resistant to chemical hydrolysis [13,14,15]. In fact, given these characteristics, PMMA has found different biomedical applications, being used for the fabrication of dental and bone cement, bone prostheses, and drug carrier systems [16,17].

Based on the previous information, different polymer matrices based on PMMA have been developed and tested to allow the entrapment of PZQ and enhance PZQ delivery, including PMMA-based nano- and microparticles [15,16,17,18]. FONSECA et al. [18,19] produced PMMA nanoparticles through mini emulsion polymerizations for load and sustain the release of PZQ, evaluating the influence of the presence of mineral oil, Eudragit^®^ E100, and the crosslinking agent ethylene glycol dimethacrylate (EGDMA) in the reacting feed. The obtained results showed that the rate of release of PZQ depended strongly on the feed composition and was very slow when pure PMMA was used as the polymer matrix. PAIVA et al. [20,21] and ALVES et al. [22] developed smart pH-responsive polymer micro- and nanoparticles [23], using 2-di(ethylamino)ethyl methacrylate (DEAEMA) and 2-di(methylamino)ethyl methacrylate (DMAEMA) as comonomers, and showing that the produced particles were pH-sensitive and that the drug release would preferentially occur in the acidic portions of the gastrointestinal tract. In fact, DMAEMA and DEAEMA polymers and copolymers have been used by others to produce materials that are sensitive to pH as a result of the ionizable pending groups that are present in the polymer chains [24].

The previously mentioned studies evaluated the performances of the produced polymer materials through in vitro dissolution tests. These dissolution studies are important for the development of new formulations because they provide cheaper alternatives for preliminary analyses of the formulation’s behavior in different media [25]. Nevertheless, for the development of new formulations, the execution of pharmacokinetic studies in animals constitutes an essential step in the preclinical stage [26]. In short, pharmacokinetics can be defined as the evaluation of drug (and respective metabolite) concentrations in peripherical and mesenteric blood after the administration of a dose. In general, the pharmacokinetics depends on several physicochemical and biopharmaceutical properties of the drug, such as its solubility and rates of dissolution in body fluids, pH sensitivity (for ionizable drugs), partition coefficient among distinct phases and tissues, concentrations, and route of administration, among others [27]. In this context, from the observed pharmacokinetic profiles, it becomes possible to determine quantitative parameters that are useful for evaluating the final performances of pharmaceutical formulations [28].

Particle size distributions can also exert significant impacts on the drug delivery profiles, affecting the drug bioavailability, targeting, release rate, and stability [29]. Small particles can enhance the rate and extent of drug absorption, leading to higher bioavailability. However, small particles can also be more susceptible to aggregation and degradation, which can negatively impact the drug’s efficacy and safety. On the other hand, large particles are generally more stable, although they usually offer higher diffusional resistance for drug release, which can reduce the rates of drug release from the particles and affect the final pharmaceutical performance of the investigated system [30].

Based on the previous paragraphs, the main objective of the present work was to evaluate the in vitro release and the in vivo pharmacokinetics of PZQ loaded to different polymer matrices, which were used for in situ PZQ entrapment. The analyzed systems presented different compositions and particle size distributions, including nanoparticles and microparticles. The obtained data indicated that all the analyzed polymer systems provided PZQ entrapment efficiencies above 95%. However, the microparticles based on PMMA and PMMA-co-DEAEMA provided higher in vitro and in vivo PZQ release profiles, allowing for a more complete release of the drug in feasible times. The obtained results indicate that, although particle size constitutes a very important parameter to facilitate and control the drug diffusion process, the composition of the polymer matrix exerted a more significant impact on the PZQ release profile in the studied system. As shown through the present manuscript, the obtained results encourage the development of a new praziquantel oral formulation that can be eventually used to treat pediatric patients more efficiently.

## 2. Materials and Methods

### 2.1. Materials

PZQ was kindly donated by Farmanguinhos (Rio de Janeiro, Brazil) in the form of a racemic mixture with a minimum assay of 99 wt%. Methyl methacrylate (MMA) (purity of 99.5 wt% and stabilized with 100 ppm of tert-butylcathecol), DEAEMA, and DMAEMA (purity of 99 wt% and stabilized with 1500 ppm of hydroquinone monomethyl ether) were provided by Sigma Aldrich (St. Louis, MO, USA). Azobisisobutyronitrile (AIBN) (purity of 99 wt% and provided by Akzo Nobel, Arnhrm, The Netherlands) was used as a free-radical initiator. Potassium persulfate (K_2_S_2_O_8_, with a purity of 99 wt%) and sodium bicarbonate (NaHCO_3_. with a purity of 99 wt%) were provided by Proquimios (Rio de Janeiro, Brazil). Hydrochloric acid (HCl), poly(vinyl alcohol) (PVA, with a minimum purity of 98 wt.%, a degree of hydrolysis of 88%, and a weight average molar mass of 2.5 × 10^4^ g/mol), sodium dodecyl sulfate (SDS), and benzoyl peroxide (BPO, with a minimum purity of 99 wt% and containing 25 wt% moisture) were provided by Vetec (Rio de Janeiro, Brazil). All chemicals were used as received, without any further purification. Ultra-pure water obtained by reverse osmosis (Gehaka, Master System MS 2000, São Paulo, Brasil) was used as the dispersion medium for polymerization reactions.

### 2.2. Preparation of Polymer Particles Loaded with PZQ

It is important to mention that PZQ presents high thermal stability, with a maximum degradation rate of around 300 °C, which ensures the stability of the drug at the reaction temperatures employed in the present work [20,21,22]. It is important to highlight that all samples were prepared and analyzed in triplicates to observe reproducibility, which was always better than 5%.

#### 2.2.1. PMMA Nanoparticles

PMMA nanoparticles loaded with PZQ were prepared through free-radical mini-emulsion polymerization, following the procedure described elsewhere [16]. The aqueous and organic phases were prepared separately, with a mass ratio of 60:40, respectively. The aqueous phase contained 1.1 wt% of SDS, 0.26 wt% of K_2_S_2_O_8,_ and 0.13 wt% of NaHCO_3_ in relation to the deionized water. In turn, the organic phase was prepared with 30 wt% of PZQ and 10 wt% of mineral oil in relation to the monomer MMA. The initial mini emulsion was obtained with the help of a high-pressure homogenizer (model APLAB-101.000 Bar, Artepeças, Taubaté, Brazil), applying a pressure of 800 bar for 10 min. The polymerization reaction was carried out in a 1-L jacketed borosilicate glass reactor, coupled to a mechanical stirrer (model Eurostar Power Control, IKA, Staufen, Germany), at 80 °C and under constant agitation of 1000 rpm for 120 min. After the predetermined time, the reaction product was cooled and stored in the form of latex for use in the in vitro and in vivo tests. The obtained particles will be referred to as PMMA-Nano.

#### 2.2.2. PMMA Microparticles

PMMA microparticles loaded with PZQ were prepared through free-radical suspension polymerization, following the procedure described elsewhere [20,21]. As described previously, the aqueous and organic phases were prepared separately, with a mass ratio of 70:30, respectively. The aqueous phase contained 1 wt% of PVA in deionized water, and the organic phase was prepared with 30 wt% of PZQ and 1 wt% of BPO in relation to monomer MMA. The polymerization reaction was carried out in a 1-L jacketed borosilicate glass reactor, coupled to a mechanical stirrer (model Eurostar Power Control, IKA, Germany), at 80 °C and under constant agitation of 1000 rpm for 120 min. After the reaction, the obtained product was washed with deionized water using a vacuum filtration system with a qualitative paper filter (80 g; porosity: 2.0 microns) and dried in an oven at 50 °C. For the in vitro and in vivo tests, microparticles with diameters smaller than 106 μm were separated using granulometric sieves equipped with an electromagnetic agitator (model VP-01, Bertel, São Paulo, Brazil). The obtained particles will be referred to as PMMA-Micro.

#### 2.2.3. Smart PMMA-co-DEAEMA and PMMA-co-DMAEMA Microparticles

The smart microparticles loaded with PZQ were prepared through free-radical suspension polymerization, following the procedure described in the literature [20,21]. The aqueous and organic phases were prepared separately, with a mass ratio of 70:30, respectively. The aqueous phase was prepared with 1 wt% of PVA in relation to deionized water, and the organic phase was prepared with 30 wt% of PZQ, 1 wt% of AIBN, and 30 wt% of DEAEMA or DMAEMA in relation to monomer MMA. The polymerization reaction was carried out in a 1-L jacketed borosilicate glass reactor, coupled to a mechanical stirrer (model Eurostar Power Control, IKA, Germany), at 80 °C and under constant agitation of 1000 rpm for 120 min. The reaction product was washed with deionized water using a vacuum filtration system with a qualitative paper filter (80 g; porosity: 2.0 micra) and dried in an oven at 50 °C. For the in vitro and in vivo tests, microparticles with diameters smaller than 106 μm were separated using granulometric sieves equipped with an electromagnetic agitator (model VP-01, Bertel, São Paulo, Brazil). The obtained particles will be referred to as PMMA-co-DEAEMA-Micro and PMMA-co-DMAEMA-Micro.

### 2.3. Characterizations

As the materials were analyzed in triplicates, the results are presented as averages of the obtained values.

The particle size distributions and polydispersity indexes (PdI) were analyzed through dynamic light scattering (Zetasizer Nano ZS 3600, Malvern Instruments, Malvern, UK, for nanoparticles; and Mastersizer 2000, Malvern Instruments, Malvern, UK, for microparticles). For nanoparticle measurements, the original samples were diluted in ultra-pure demineralized water (1:500 by volume). For size distribution measurements, the particles were dispersed in distilled water to be placed in the equipment compartment. The presence of an agitation system guaranteed the dispersion and homogenization of the particles in the medium.

The entrapment efficiency was determined through UV–VIS spectrometry (model Multiskan Go, ThermoScientific, Vantaa, Finland). In order to do that, the polymer microparticles were weighed, dispersed in methanol, and left under magnetic stirring for 24 h. The material was filtrated using a syringe filter, Chromafil XTRA PET (25 mm–0.20 μm), and the supernatant was analyzed through UV–VIS at 263 nm. As nanoparticles were stored in the form of latex (a direct product of the reaction), they were previously dried in an oven at 50 °C, dispersed in methanol, filtrated with Amicon membranes, and analyzed at the same wavelength. Regarding the quantification, an analytical curve was constructed with PZQ in methanol, and all samples were measured in triplicates. To remove background interference from the polymer matrices, the absorbance of blanks (particles without the drug) was subtracted.

In vitro dissolution evaluation was performed using a dissolution tester (Ethik Technology, São Paulo, Brazil) to verify the release rates of PZQ. The media comprised an aqueous HCl 0.1 M solution containing 0.2 wt% sodium lauryl sulfate, in accordance with the methodology described by the Brazilian Pharmacopeia for PZQ tablets [31]. The tests were conducted in triplicates using USP apparatus 2, operated at 37 °C under constant stirring of 50 rpm for a period of 3 h. For the in vitro dissolution assays, amounts of micro/nanoparticles corresponding to 0.66 g/L of PZQ were used. It is noteworthy that the evaluated microparticles and nanoparticles were added directly into the dissolution medium, without the aid of any capsule or membrane.

To model the release profiles, the Excel plugin DDSolver was used [32]. The tested kinetic models were the zero-order, first-order, Higuchi, Hixson–Crowell, Korsmeyer–Peppas, Hopfenberg, and Peppas–Sahlin models [33].

The in vivo tests were carried out by CIEnP (Centro de Inovação e Ensaios Pré-Clínicos), which specializes in pharmacokinetic studies, and followed the rules of the ethical principles in animal experimentation adopted by the National Council for the Control of Animal Experimentation (CONCEA). The protocol used was reviewed and approved by the committee on ethics in animal use (CEUA) of CIEnP under the numbers 217/01 and addenda 001 and 264/00 and is filed at CIEnP. In order to do that, healthy rats of the *Rattus norvegicus* species and Sprague Dawley lineage from the breeding vivarium of CIEnP were used. The animals (whose matrices were purchased from Charles River, USA) were kept under SPF (specific pathogen-free) conditions. The samples were prepared in glass vials, using 2 wt% of Cremophor in an aqueous solution as a vehicle, at a final concentration of 60 mg/mL. The use of Cremophor is recommended to prevent particle agglomeration inside the thin cannula used to perform the oral administration of the doses to the lab rats. The nanoparticles were suspended at a concentration of 52 mg/mL, used neat, and the volume was adjusted at the time of administering the animals to match the concentration of the other test items. The administration was by oral route (gavage), and the dose used was 60 mg/kg.

## 3. Results and Discussion

### 3.1. Particle Size Distributions

As indicated previously, in the present work, particles of different sizes and compositions were used for PZQ entrapment. It is well known that particle size distributions can exert a significant effect on the drug delivery profile, affecting the drug’s bioavailability, targeting, release rate, and stability [29].

Figure 1A and Table 1 show the size distributions and average diameters of the different analyzed particles. As shown, the produced microparticles exhibited size distributions in the micrometric range, as expected, as a consequence of using the suspension polymerization technique for particle manufacture [34]. The average diameters ranged between 27 and 180 µm. It can also be observed that the DMAEMA copolymer microparticles presented a particle size distribution that was similar to that of pure PMMA, although slightly wider. In turn, the size distribution of the copolymer produced with DEAEMA was shifted towards smaller diameters. The production of cationic copolymers based on DEAEMA and DMAEMA at different concentrations was studied by PAIVA et al. [21]. These authors also reported the formation of smaller particle sizes with DEAEMA when compared with DMAEMA, as was also observed in the present study. As a whole, it can be observed that the presence of the selected comonomers did not affect the formation of polymer microparticles, although they had distinct size distributions.

For the desired drug delivery application, smaller particle sizes are believed to be better, due to the less pronounced diffusional resistance and consequently higher drug delivery efficiency [35,36]. It must be pointed out that the adjustment of microparticle sizes can be achieved through the manipulation of reaction operation parameters, such as agitation, type of surfactant, and the ratio between the dispersed and continuous phases, which in turn can also enable the manipulation of drug delivery rates [37]. In the present work, the selection of the smallest particle sizes was carried out with the help of a vibrating sieve, with the selection of microparticles with diameters smaller than 106 μm, as recommended by PAIVA et al. [21]. After sieving, the PMMA-co-DEAEMA product displayed the smallest average diameter of 29 μm, although with a high polydispersity index. A wide particle size distribution can affect the release profile, as particles of different sizes will probably result in different rates of diffusion and sedimentation, which can lead to uneven distribution and localization of the drug in the body [38]. Additionally, wide size distributions can also affect the rate of drug release, as smaller particles present a higher surface-to-volume ratio and can release the drug more quickly [39].

Figure 1B presents the particle size distributions of the produced nanoparticles, showing a monomodal size distribution with an average diameter of around 170 nm. It is important to highlight that similar nanoparticles were studied previously by others [40], who indicated that these emulsions can be stored stably at rest for at least 6 months at room temperature.

### 3.2. Entrapment Efficiency

As presented in Table 2, the results indicate that all polymer systems (micro and nanoparticles) provided high PZQ entrapment efficiencies, near 100 %. Particularly, it is noteworthy that praziquantel presents low solubility in water [7], which favors its migration and, consequently, its loading in the organic phase. In this context, the slight reduction in entrapment efficiency of smart microparticles can be explained by the increase in hydrophilicity of the organic phase as a result of the presence of hydrophilic cationic comonomers [41]. In general, the obtained results are consistent with the literature [18,20,21,22].

### 3.3. Dissolution

Figure 2 shows the release profiles of PZQ from the different polymer matrices. As expected, the protonation of the comonomers at lower pH values led to swelling of the particles and faster release of the drug, as reported previously for other microscopically dispersed systems [6,22,42,43,44]. It is important to mention that the smart microparticles exhibited PZQ release rates that were higher than the ones observed for PMMA nanoparticles, emphasizing the existence of chemical and physicochemical characteristics that can balance the presumably higher diffusional resistance offered by the larger microparticles. In the case of smart copolymers, the release mechanism involves two simultaneous steps. First, the particle is swollen with water in acidic stomach conditions, reducing the diffusion resistance for drug release. Then, the drug diffuses into the bulk acidic aqueous medium, where it is absorbed by the body. When compared to the other analyzed polymer matrices, the much lower diffusion resistance offered by the swollen smart particles can explain the better performances in terms of PZQ concentrations in the plasma. When compared to the free PZQ form, the amorphous nature of the PZQ molecules dissolved in the smart polymer particles [20,21,22] can explain the faster and more efficient release of PZQ in the body. In this case, the smart microparticles based on PDMAEMA and PDEAEMA are biocompatible cationic polymers characterized by the presence of tertiary amines in their side chains, which are characterized by pH- sensitivity. In short, pH-sensitive polymers are characterized by the presence of ionizable groups in their structures, presenting physicochemical properties dependent on this variable. These groups can be acidic (for example, carboxylic acid) or basic (for example, ammonium ions), which can accept or release protons in the face of changes in pH, so that, when they are charged, they generate electrostatic repulsions that allow an increase in the hydrodynamic volume of chains and polymer solubility [42,45].

It is particularly important to note that the PZQ release rates observed for PMMA nanoparticles were significantly smaller than the ones observed for PMMA microparticles. This unexpected result was also reported previously by ALVES et al. [22] and associated with the possible accumulation of charges on the surfaces of the suspended nanoparticles, since the accumulation of charges at the interface can provide additional resistance for mass transfer between the organic phase and the aqueous medium, preventing the drug release. This constitutes an important issue for those who work in the field, as it has been normally accepted that nanoparticles will necessarily provide faster rates of drug release than microparticles, although the results presented here clearly indicate that other particle properties and characteristics can deny this usual expectation.

Table 3 shows the R-squared (R^2^) and mean squared error (MSE) for the models built with the DDSolver plugin. In zero-order kinetics, the drug release is only a function of time, and the process takes place at a constant rate independent of drug concentration. In the first-order model, the release is dependent only on concentration; the Higuchi model describes the drug diffusion process based on Fick’s law, depending on the square root of time; Hixson-Crowell states that the drug release is not by diffusion but rather limited by dissolution velocity, which can occur through the polymeric matrix; the Hopfenberg model is used to explain the release of active agents from erodible polymer matrices; the Korsmeyer-Peppas and Peppas-Sahlin models will be explained in more detail below since these were the ones that best represented the drug release in this work [33]. The PZQ release profile from PMMA microparticles did not provide very good fits with any of the tested models, showing R^2^ values around 0.7 and very high MSE for all models. This can possibly be explained by the fast stabilization of the release profile, suggesting the fast release of PZQ molecules placed near the particle surface and the very slow release of PZQ molecules located inside the particle due to strong diffusion constraints. On the other hand, the Korsmeyer-Peppas model and the Peppas-Sahlin models seemed to promote better fit of experimental data for smart microparticles. However, it can be noted that while the PMMA-co-DEAEMA matrix provided the best fit with the Korsmeyer-Peppas model (R^2^ = 0.991), the PMMA-co-DMAEMA matrix exhibited the best fit with the Peppas-Sahlin model (R^2^ = 0.964) (Figure 3). The obtained responses suggest that all synthesized microparticles can be subject to different release mechanisms and that pH sensitivity can be of fundamental importance for the desired application.

The Korsmeyer-Peppas model is believed to be useful when the release mechanism is unknown or when more than one type of drug release phenomenon is involved. It is represented by two parameters: a diffusion coefficient (n) and a release constant (Kkp) that can be seen in Table 4 [46]. The obtained results for the Korsmeyer-Peppas model indicated that the diffusion coefficients (n) of PZQ were smaller than 0.5 for both smart microparticles, indicating that the diffusion mechanisms of PZQ in these systems were Fickian. For the PMMA microparticles, the diffusion coefficient (n) was equal to 0.6, which indicates that the model predicts the occurrence of anomalous transport or non-Fickian diffusion and that the mechanism of PZQ release is governed by diffusion and swelling [47,48]. Regarding the release constant (Kkp), the rates of drug release increase with the parameter values. The matrix produced with DEAEMA presented the highest Kkp value (Kkp = 53.9), followed by the matrix prepared with DMAEMA (Kkp = 22.8) and PMMA (Kkp = 0.8), confirming the profiles observed in Figure 2.

In turn, the Peppas-Sahlin model is characterized by the premise that the release mechanism is controlled by two mechanisms: diffusion and polymer relaxation. This model contains three parameters, that can be seen in Table 4, with the diffusional contribution represented by constant kd, the relaxational contribution represented by constant kr, and an additional shape parameter represented by *m* [33,49]. In general, the obtained results indicated the prevalence of Fickian diffusion and a small contribution from chain relaxation (kd >> kr). Indeed, both smart microparticles presented negative kr values, suggesting the short duration of the Fickian diffusion approximation for spheres [48,49]. Furthermore, both polymer matrices showed *m* values of 0.45, indicating that the release mechanism follows mainly Fickian-type diffusion [50].

Although the PZQ release rates from nanoparticles were close to zero, the obtained release profile was also modeled, but satisfactory values were not found.

### 3.4. Pharmacokinetic Studies

Based on the drug’s pharmacokinetic profile, it is possible to obtain some quantitative parameters that are useful for evaluating the performance and comparing the behavior of different formulations [51]. In general, the obtained results showed that encapsulated PZQ was released from the different polymer systems and absorbed in the gastrointestinal tract after oral administration. The PZQ plasma concentration profiles in the different evaluated groups are shown in Figure 4, and their respective pharmacokinetic parameters are listed in Table 5.

The results presented in Figure 4 shows that, after 10 min of oral administration, it was already possible to detect the presence of PZQ in the samples, even when PZQ had been encapsulated, and it continued to be observed after 20 h of testing. Furthermore, it is possible to observe a clear difference between the different evaluated systems. As observed, PZQ in its free form reached its maximum concentration (near 400 ng/L) after 15 min of oral administration, reaching higher plasma concentrations in the case of PMMA microparticles and similar to those observed with PMMA nanoparticles. Thus, the results seem to indicate that the drug release process through these systems was, in some way, impaired, probably being dependent on the diffusion process through the polymer matrix. In this context, it is important to observe that the PMMA microparticles presented a delayed release effect, reaching their maximum plasma concentrations in relatively longer times than the nanoparticles, confirming the diffusional resistance through the larger average diameters in the living organisms and in contact with the actual body fluids. Such results seem to emphasize that particle size is an extremely important parameter that must be taken into account with regard to the release process in the gastrointestinal tract [50].

Indeed, Durrer et al. [52] showed that the size of polystyrene nanoparticles seemed to significantly influence the absorption process in the intestinal mucosa of rats, where nanoparticles with diameters of 230 nm reached equilibrium after 10 min of testing, while nanoparticles with diameters of 670 nm only reached the equilibrium after 30 min. DESAI et al. [53] confirmed the importance of size by showing that particles with smaller diameters (0.1 μm) provided higher absorption efficiencies than larger ones (10 μm).

On the other hand, Figure 4 shows that smart microparticles based on PMMA-co-DEAEMA achieved significantly higher plasma concentrations (C_max_ = 1007 ± 83 ng/mL) than free PZQ and other systems, indicating the increased rate of release of PZQ and, thus, confirming the promising potential of using these polymer materials for drug entrapment and delivery. These results show that the release process is governed simultaneously by at least two factors: the size of the particles and the composition of the polymer matrix, as also reported by Delie and Blanco-Príeto et al. [54]. These results also indicate that the presence of the cationic comonomer positively influenced the performance of the smart microparticles. Additionally, bioavailability can also be evaluated through the area under the curves (AUC), which were statistically similar for free PZQ, PZQ encapsulated in nanoparticles and PZQ encapsulated in PMMA-co-DEAEMA microparticles, calculated with the trapezoidal rule 8 h after the administration. This reinforces the consistency of the proposed analyses. Calculations of the AUC for longer periods of time were not precise and not used for comparison purposes, given the low attained concentration values (which lead to low signal/noise ratio) and long last sampling interval, which must be considered in future studies.

These results corroborate those presented previously during the in vitro release studies, as the polymer matrix prepared with DEAEMA provided the best performance for the PZQ release. This may be associated with the ability of this matrix to undergo protonation when in contact with the acid pH of the stomach, allowing the matrix to swell. This swelling enhances the diffusion process of the drug through the matrix, as observed in the in vitro release study modeled by Korsmeyer-Peppas. In addition, the particle size was also smaller than for the other microparticles, as observed in the DLS analysis, which contributes to the faster release of the drug since the smaller sizes lead to higher surface areas.

This study confirms the promising potential of using polymer microparticles in the pharmaceutical industry. As a matter of fact, polymer microparticles can be used to improve the bioavailability of drugs and provide means to adjust the respective administration forms, constituting an important area of research for the development of new and improved drug formulations. Moreover, microparticles can be utilized to make medicine administration easier, especially for children. In the specific case of praziquantel, which has a bitter taste and must be administered in relatively high dosages, microparticles can be formulated into liquid or semi-solid forms to mask the drug’s taste, enhance the drug’s efficiency, and make drug administration easier. This is particularly important for pediatric patients, who may have difficulties swallowing large pills and may demand fractionation of dosages. With the help of microparticles, drug delivery can be customized to meet the specific needs of patients, thereby improving treatment outcomes and patient compliance.

## 4. Conclusions

The present study evaluated the in vitro and in vivo release rates of encapsulated praziquantel (PZQ) from different polymer matrices based on poly(methyl methacrylate) (PMMA) and cationic comonomers (2-(diethylamino)ethyl methacrylate) (DEAEMA) and (2-(dimethylamino)ethyl methacrylate) (DMAEMA) for the first time. The employed polymerization techniques were efficient in producing polymer nano- and microparticles, allowing the entrapment of PZQ with efficiencies above 95%. The obtained results showed that the PZQ entrapment in pH-sensitive PMMA-co-DEAEMA particles can promote many competitive advantages to the treatment of schistosomiasis since this matrix displayed better release profiles, reaching values close to 100% of PZQ release in less than 3 h. The in vivo pharmacokinetic analyses were conducted using free PZQ as the control group that was compared with the investigated matrices. This study also showed that microspheres based on DEAEMA provided the best absorption profile, displaying a Cmax average value (1007 ± 83 ng/mL) that was 2.2 times higher than that obtained for PZQ in the free form (432 ± 98 ng/mL). Based on all these positive and encouraging results, the polymer matrix based on PMMA-co-DEAEMA was selected as the best matrix among the investigated ones for the preparation of a new pharmaceutical product.

## Figures and Tables

**Figure 1 materials-16-03382-f001:**
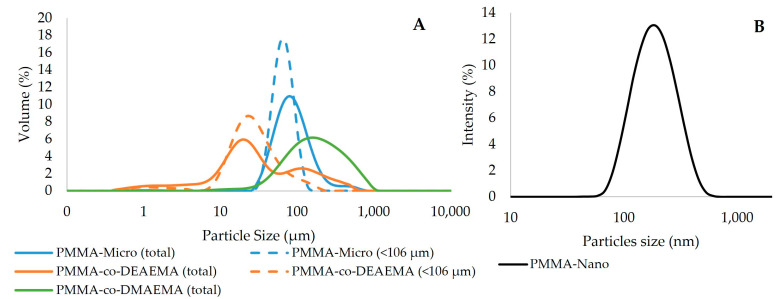
(**A**) Particle size distributions of the produced polymer microparticles. (**B**) Particle size distribution of the produced polymer nanoparticles.

**Figure 2 materials-16-03382-f002:**
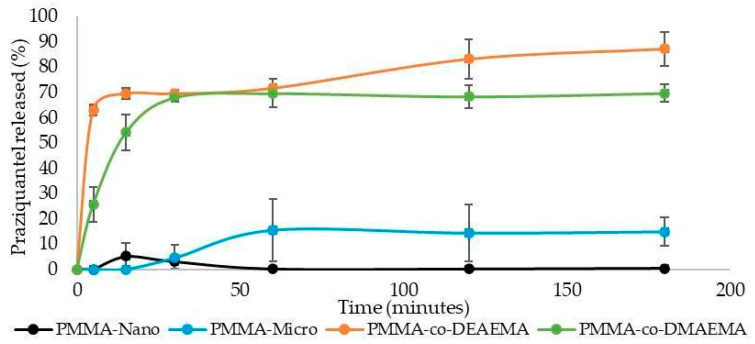
PZQ release profiles from the produced polymer micro- and nanoparticles.

**Figure 3 materials-16-03382-f003:**
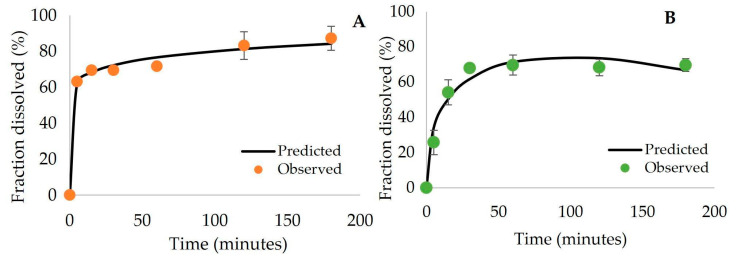
(**A**) Observed and predicted PZQ release profiles for PMMA-co-DEAEMA microparticles with the Korsmeyer-Peppas model. (**B**) Observed and predicted PZQ release profiles for PMMA-co-DMAEMA microparticles with the Peppas-Sahlin model.

**Figure 4 materials-16-03382-f004:**
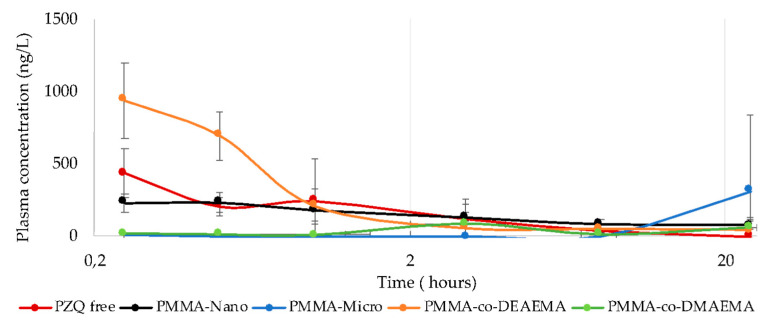
PZQ plasma concentration profiles after oral administration of the following groups: free-PZQ, PMMA-Nano, PMMA-Micro, PMMA-co-DEAEMA, and PMMA-co-DMAEMA.

**Table 1 materials-16-03382-t001:** Average sizes of produced micro- and nanoparticles.

Sample	Average Particle Size (Total)	SPAN (Total) *	Average Particle Size (<106 μm)	SPAN (<106 μm)
PMMA-Nano	170 ± 1.3 nm	0.90	-	-
PMMA-Micro	89.9 ± 0.2 μm	1.5	69.4 ± 0.7 μm	0.78
PMMA-co-DEAEMA	27 ± 1 μm	7.2	28.7 ± 3 μm	2.3
PMMA-co-DMAEMA	173 ± 11 μm	2.6	63.7 ± 0.6 μm	1.3

* SPAN = (d90 − d10)/d50.

**Table 2 materials-16-03382-t002:** PZQ entrapment efficiencies in the produced nano- and microparticles.

Sample	Concentration (mg/mL)	Entrapment Efficiency (%)
PMMA-Nano	0.60 ± 0.02	100 ± 4
PMMA-Micro	0.80 ± 0.10	100 ± 9
PMMA-co-DEAEMA	0.58 ± 0.03	96 ± 6
PMMA-co-DMAEMA	0.60 ± 0.01	99 ± 2

**Table 3 materials-16-03382-t003:** Model performances for the PZQ release profiles.

Model	Sample	R^2^	MSE
Zero-order	PMMA-Micro	0.68	18
	PMMA-co-DEAEMA	−1.27	1941
	PMMA-co-DMAEMA	−0.65	1250
First-order	PMMA-Micro	0.71	16
	PMMA-co-DEAEMA	−0.05	893
	PMMA-co-DMAEMA	0.20	603
Higuchi	PMMA-Micro	0.78	12
	PMMA-co-DEAEMA	0.09	776
	PMMA-co-DMAEMA	0.42	435
Korsmeyer-Peppas	PMMA-Micro	0.79	1
	PMMA-co-DEAEMA	0.99	9
	PMMA-co-DMAEMA	0.86	128
Hixson-Crowell	PMMA-Micro	0.70	17
	PMMA-co-DEAEMA	−0.28	1090
	PMMA-co-DMAEMA	0.01	748
Hopfenberg	PMMA-Micro	0.70	20
	PMMA-co-DEAEMA	−0.28	1308
	PMMA-co-DMAEMA	0.05	866
Peppas-Sahlin	PMMA-Micro	0.71	24
	PMMA-co-DEAEMA	0.83	220
	PMMA-co-DMAEMA	0.96	40

**Table 4 materials-16-03382-t004:** Obtained parameters of the Korsmeyer-Peppas model and the Peppas-Sahlin model for the polymer microparticles.

Model	Parameter	PMMA-co-DEAEMA	PMMA-co- DMAEMA	PMMA_Microo_
Korsmeyer-Peppas	Kkp	54	23	0.77
	n	0.09	0.25	0.61
Peppas-Sahlin	kd	23	19	2.18
	kr	−1.5	−1.2	−0.06
	m	0.45	0.45	0.45

**Table 5 materials-16-03382-t005:** Pharmacokinetic parameters of animals treated with the evaluated polymer matrices (confidence level of 95%).

Parameters ^1^	PZQ Free	PMMA_Nano_	PMMA_Micro_	PMMA-co-DEAEMA	PMMA-co-DMAEMA
C_max_ (ng/mL)	432 ± 98	286 ± 14	42 ± 19	1007 ± 83	134 ± 59
AUC_last_ ^2^ (h*ng/mL)	1012 ± 424	1012 ± 166	2 ± 9	1063 ± 390	320 ± 94
AUC_INF_obs_ ^3^ (h*ng/mL)	1302 ± 525	5341 ± 1386	-	1677 ± 384	-
K_e_ (1/h) ^3^	0.25 ± 0.08	0.03 ± 0.01	-	0.34 ± 0.08	-
T_max_ (h)	0.25	0.41 ± 0.08	8	0.33 ± 0.08	17.00 ± 7.00
T_1/2β_ (h) ^3^	3.64 ± 1.46	6.20 ± 1.96	-	2.24 ± 0.46	-

^1^ C_max_ (maximum concentration); AUC (area under the curve); K_e_ (elimination constant); T_max_ (time to reach maximum concentration); T_1/2β_ (elimination half-life). Data expressed as mean ± confidence interval of mean. ^2^ Last point collected after 8 h. ^3^ Calculated with help of an exponential interpolator.

## Data Availability

The data that support the findings of this study are available from the corresponding author upon reasonable request.

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
