# Peer review of "In Vitro Release and In Vivo Pharmacokinetics of Praziquantel Loaded in Different Polymer Particles"

_materials, 2023, doi:10.3390/ma16093382_

Round 1
Reviewer 1 Report
The authors performed in vitro and in vivo studies of praziquantel micro/nanoparticles. Even though the study is properly designed, some of the methodologies are not clear.
The in vitro dissolution test procedure is not clear. Is the microparticle directly put into the medium or filled in a capsule or cellophane membrane?
Figure 1 B is not a representative figure for the size distribution, because the size in the text is mentioned as 106, but the figure represents a size between 400 to 500 nm.
the pharmacokinetic study protocol the sample preparation is not clear. the authors state that ....."The samples were prepared in glass 220 vials, using 2 wt% of cremophor in aqueous solution as a vehicle, at a final concentration of 60 mg/mL. The nanoparticles were suspended at a concentration of 52 mg/mL, used neat and the volume was adjusted at the time of administering the animals to match the concentration of the other test Items.....
why cremophor is used? cremophor may influence the solubility of drug
Author Response
Answers to Reviewer#1's Questions
Comments and Suggestions for Authors:
Q1: The authors performed in vitro and in vivo studies of praziquantel micro/nanoparticles. Even though the study is properly designed, some of the methodologies are not clear.
A1: We thank Reviewer#1 for his/her encouraging initial remarks and hope Reviewer#1 will appreciate this new version of the manuscript better.
Q2: The in vitro dissolution test procedure is not clear. Is the microparticle directly put into the medium or filled in a capsule or cellophane membrane?
A2: We sincerely understand Reviewer#1's concerns, and thank him/her for the feedback. To clarify this point, both microparticles and nanoparticles were added directly into the dissolution medium, without using any sort of capsule or membrane. Such information was added to this revised version of the manuscript.
Q3: Figure 1 B is not a representative figure for the size distribution, because the size in the text is mentioned as 106, but the figure represents a size between 400 to 500 nm.
A3: We thank Reviewer#1 for his/her comment regarding Figure 1B of our manuscript. We would like to clarify that the size distribution of the nanoparticles displayed in the figure is representative of the nanosized samples that were analyzed using dynamic light scattering (DLS). These samples were neither dried nor sieved and were taken directly for size analysis after the reaction. The size distribution of these particles ranged from 40 to 600 nm with an average of 170 nm, as shown in Table 1. Regarding the value of 106 μm, this refers to the size of the microsized samples that were sieved with an electromagnetic sieve. The purpose of sieving was to separate particles smaller than 106 μm for further analyses. We hope that this explanation can clarify the average sizes described in our study.
Q4: The pharmacokinetic study protocol the sample preparation is not clear. The authors state that ....."The samples were prepared in glass 220 vials, using 2 wt% of cremophor in aqueous solution as a vehicle, at a final concentration of 60 mg/mL. The nanoparticles were suspended at a concentration of 52 mg/mL, used neat and the volume was adjusted at the time of administering the animals to match the concentration of the other test Items”
Why is cremophor used? cremophor may influence the solubility of drug.
A4: We sincerely understand Reviewer#1's concerns and thank him/her for the feedback. The use of cremophor is recommended to prevent particle agglomeration inside the thin cannula used to perform the oral administration of the doses to the lab rats. This is in accordance with the standard experimental procedures, as there is no empirical evidence that the used cremophor concentrations can affect the significance of the obtained data, particularly because the suspended particles are solid. Such information was added to this revised version of the manuscript.

Reviewer 2 Report
1. The Conclusion section should be clear and concise: the important results and main conclusions drawn in this paper should be highlighted and presented in more precise language.
2. The manuscript lacks information on experimental replication. This is particularly worrisome. Please revise the manuscript detailing your experimental and technical replications (refering 10.1021/acsami.1c25014).
3. Where are these drug-loaded particles going to be used in real life? The novelty of this work can be described at the end of the Introduction. It could be better if a brief comment (challenges and future prospects) is added at the manuscript.
4. Mechanism descriptions for drug delivery can be strengthened by citing 10.1016/j.reactfunctpolym.2020.104501 and what are the advantages of the current work compared to published articles?
5. There are some formatting errors in the article. For example, spelling of references must be checked to meet the journal style (such as Reference 9). Please check carefully and use abbreviation properly.
Author Response
Answers to Reviewer#2's Questions
Comments and Suggestions for Authors:
Q1: The Conclusion section should be clear and concise: the important results and main conclusions drawn in this paper should be highlighted and presented in more precise language.
A1: We sincerely understand Reviewer#2's concerns. As requested, at the Conclusion section of the revised manuscript one can now read:
“The present study evaluated in vitro and in vivo release rates of encapsulated praziquantel (PZQ) from different polymer matrices based on poly(methyl methacrylate) (PMMA) and cationic comonomers (2-(diethylamino)ethyl methacrylate) (DEAEMA) and (2-(dimethylamino)ethyl methacrylate) (DMAEMA) for the first time. The employed polymerization techniques were efficient to produce polymer nano- and microparticles, allowing the encapsulation of PZQ with efficiencies above 95 %.
The obtained results showed that the PZQ encapsulation in pH-sensitive PMMA-co-DEAEMA particles can promote many competitive advantages to the treatment of schistosomiasis, since this matrix displayed better release profiles reaching values close to 100 % of PZQ release in less than 3 h. The in vivo pharmacokinetic analyses were conducted using free PZQ as the control group that was compared with the investigated matrices. This study also showed that microspheres based on DEAEMA provided the best absorption profile, displaying Cmax average value (1007 ± 83 ng/mL) that was 2.2 times higher than obtained for PZQ in the free form (432 ± 98 ng/mL). Based on all these positive and encouraging results, the polymer matrix based on PMMA-co-DEAEMA was selected as the best matrix among the investigated ones for preparation of a new pharmaceutical product.”
We hope Reviewer#2 will appreciate this revised version of the manuscript better.
Q2: The manuscript lacks information on experimental replication. This is particularly worrisome. Please revise the manuscript detailing your experimental and technical replications (refering 10.1021/acsami.1c25014).
A2: We thank Reviewer#2 for the feedback and apologize for not providing this information in the first version of the manuscript. We would like to clarify that all experiments were performed and analyzed in triplicates and proved to be reproducible.
At the Materials and Methods section of the revised manuscript, one can now read:
"It is important to highlight that all samples were prepared and analyzed in triplicates to observe reproducibility, which was always better than 5%."
Besides, one can also read:
"As the materials were analyzed in triplicates, results are presented as averages of the obtained values."
We hope this can clarify this important point raised by Reviewer#2.
Q3: Where are these drug-loaded particles going to be used in real life? The novelty of this work can be described at the end of the Introduction. It could be better if a brief comment (challenges and future prospects) is added at the manuscript.
A3: We thank Reviewer#2 for this important remark. As requested, at the end of the Introduction section one can read now:
"As shown through the present manuscript, the obtained results encourage the development of a new praziquantel oral formulation that can be eventually used to treat pediatric patients more efficiently."
We hope Reviewer#2 will appreciate this revised version of the manuscript better.
Q4: Mechanism descriptions for drug delivery can be strengthened by citing 10.1016/j.reactfunctpolym.2020.104501 and what are the advantages of the current work compared to published articles?
A4: We sincerely understand Reviewer#2's concerns. As suggested, we added a reference to the article "Recent advances in natural polymer-based drug delivery systems" by Tong et al. (2020), published in Reactive and Functional Polymers, and strengthened the description of the drug delivery mechanism.
Regarding the advantages of our work when compared to other published articles, we believe that our study offers several unique contributions to the field. For instance, the present article shows for the first time that the use of the PMMA-co-DEAEMA smart matrix can provide praziquantel plasmatic values that are ​​higher than the values obtained with the free drug, thus improving the biodisponibility of the drug. As the in vivo experiments were certified by the National Health Surveillance Agency of Brazil (ANVISA), these results can be useful for certification of a commercial product in the near future, which is indeed in progress.
Based on the previous comments, at the end of the Results section of this revised version of the manuscript one can read now:
“This study confirms the promising potential of using polymer microparticles in the pharmaceutical industry. As a matter of fact, polymer microparticles can be used to improve the bioavailability of drugs and provide means to adjust the respective administration forms, constituting an important area of research for development of new and improved drug formulations. Moreover, microparticles can be utilized to turn medicine administration easier, especially for children. In the specific case of praziquantel, which has a bitter taste and must be administered in relatively high dosages, microparticles can be formulated into liquid or semi-solid forms to mask the drug taste, enhance the drug efficiency and make drug administration easier. This is particularly important for pediatric patients, who may face difficulties to swallow large pills and may demand fractionation of dosages. With help of microparticles, drug delivery can be customized to meet the specific needs of patients, thereby improving treatment outcomes and patient compliance.”
Q5. There are some formatting errors in the article. For example, spelling of references must be checked to meet the journal style (such as Reference 9). Please check carefully and use abbreviations properly.
A5: We thank Reviewer#2 for this remark. The manuscript was revised, as requested.

Reviewer 3 Report
This manuscript, entitled "In Vitro Release and In Vivo Pharmacokinetics of Praziquantel Loaded in Different Polymer Particles," presents an evaluation of the in vitro release and in vivo pharmacokinetics of praziquantel loaded in different polymer particles by Emiliane Daher Pereira, Luciana da Silva Dutra, Thamiris Franckini Paiva, Larissa Leite de Almeida Carvalho, Helvécio Vinícius Antunes Rocha, and José Carlos Pinto. The authors used poly(methyl methacrylate) (PMMA) nano- and microparticles and PMMA-co-DEAEMA (2-(diethylamino)ethyl methacrylate) and PMMA-co-DMAEMA (2-(dimethylamino)ethyl methacrylate) microparticles for the encapsulation of PZQ. The results showed that the PMMA-co-DEAEMA matrix had the best release profile, reaching almost 100% of PZQ released in 3 hours of study. The pharmacokinetic study also showed that microspheres with DEAEMA had the best absorption profile, with a Cmax average 2.2 times higher than PZQ in the free form. The authors concluded that the size of the particles and the composition of the polymer matrix are important factors that influence the release and absorption of PZQ.
This is a well-written and comprehensive manuscript that provides valuable insights into the release and absorption of PZQ in different polymer particles. However, some minor revisions are suggested. For example, the authors should provide more details on the particle size distributions and average diameters of the different analyzed particles. Additionally, the authors should provide more information on the models used to analyze the release profiles, such as the Korsmeyer-Peppas and Peppas-Sahlin models. There is a dearth of references to relevant works on this topic for instance, supercritical carbon dioxide could be cited as an additional method for loading drugs into a semi-dimensional PMMA matrix. This could emphasize the increasing interest in this area and the relevance of this study.
References: [1] https://doi.org/10.3390/polym14235332; [2] https://doi.org/10.1134/S1990793121070101; [3] https://doi.org/10.1016/j.jcou.2021.101553 Finally, the authors should discuss the implications of their findings for the development of drug delivery systems for praziquantel.
Author Response
Answers to Reviewer#3's Questions
Comments and Suggestions for Authors
Q1: This manuscript, entitled "In Vitro Release and In Vivo Pharmacokinetics of Praziquantel Loaded in Different Polymer Particles," presents an evaluation of the in vitro release and in vivo pharmacokinetics of praziquantel loaded in different polymer particles by Emiliane Daher Pereira, Luciana da Silva Dutra, Thamiris Franckini Paiva, Larissa Leite de Almeida Carvalho, Helvécio Vinícius Antunes Rocha, and José Carlos Pinto. The authors used poly(methyl methacrylate) (PMMA) nano- and microparticles and PMMA-co-DEAEMA (2-(diethylamino)ethyl methacrylate) and PMMA-co-DMAEMA (2-(dimethylamino)ethyl methacrylate) microparticles for the encapsulation of PZQ. The results showed that the PMMA-co-DEAEMA matrix had the best release profile, reaching almost 100% of PZQ released in 3 hours of study. The pharmacokinetic study also showed that microspheres with DEAEMA had the best absorption profile, with a Cmax average 2.2 times higher than PZQ in the free form. The authors concluded that the size of the particles and the composition of the polymer matrix are important factors that influence the release and absorption of PZQ. This is a well-written and comprehensive manuscript that provides valuable insights into the release and absorption of PZQ in different polymer particles. However, some minor revisions are suggested.
A1: We sincerely thank Reviewer#3 for his/her encouraging initial remarks and hope Reviewer#3 will appreciate this revised version of the manuscript better.
Q2: For example, the authors should provide more details on the particle size distributions and average diameters of the different analyzed particles.
A2: We thank Reviewer#3 for this remark. As requested, we provide more details on these important parameters.
In our study, particles of different sizes and compositions were used for PZQ encapsulation, and we presented the size distributions and average diameters of the analyzed particles in Figure 1 and Table 1. As discussed in the manuscript, smaller particle sizes are believed to be better for drug delivery due to the less pronounced diffusional resistance and higher drug delivery efficiency. The selection of the smallest particle sizes was carried out with the help of a vibrating sieve, with the selection of microparticles with diameters smaller than 106 μm, as recommended by previous studies. A wide particle size distribution can affect the release profile, and we discussed this topic in detail in the manuscript. We also discussed how wide size distributions can affect the rate of drug release, as smaller particles present a higher surface-to-volume ratio and can release the drug more quickly. Based on the previous remarks, ee believe that the information on particle size distributions and their effects on drug delivery profiles are very important and are now discussed more deeply in this revised version of the manuscript.
Q3: Additionally, the authors should provide more information on the models used to analyze the release profiles, such as the Korsmeyer-Peppas and Peppas-Sahlin models.
A3: We thank Reviewer#3 for this remark. As requested, we have now included the required information into the Dissolution part of the Results and Discussion section.
"According to the zero-order kinetics, the drug release is only a function of time and the process takes place at a constant rate that is independent of the drug concentration. According to the first-order rate model, the release is dependent only on concentration. The Higuchi model describes the drug diffusion process based on the Fick's law, depending on the square root of time. On the other hand, the Hixson-Crowell model assumes that the drug release is not controlled by diffusion, but rather limited by the dissolution velocity in the bulk medium. The Hopfenberg model is normally used to explain the release of active agents from erodible polymer matrices. Finally, the Korsmeyer-Peppas and Peppas-Sahlin models will be explained in more detail below because these models provided the best fits to represent the rates of drug release in the present work."
Q4: There is a dearth of references to relevant works on this topic for instance, supercritical carbon dioxide could be cited as an additional method for loading drugs into a semi-dimensional PMMA matrix. This could emphasize the increasing interest in this area and the relevance of this study.
References:
[1]https://doi.org/10.3390/polym14235332;
[2] https://doi.org/10.1134/S1990793121070101;
[3] https://doi.org/10.1016/j.jcou.2021.101553.
A4: We sincerely understand Reviewer#3's concerns and thank him/her for this comment. As suggested, references to works on loading drugs into a semi-dimensional PMMA matrix using supercritical carbon dioxide were included into this revised version of the manuscript. At the Introduction section of the manuscript one can read now:
“In addition to these traditional methods, new techniques based on use of supercritical CO2 have been reported in the literature as promising alternatives for replacing commonly used toxic organic solvents in release applications. In this scenario, supercritical CO2 technologies can be considered as cleaner and more sustainable strategies for the encapsulation of several active principles.”
We hope Reviewer#3 will appreciate this revised version of the manuscript better.
Q5: Finally, the authors should discuss the implications of their findings for the development of drug delivery systems for praziquantel.
A5: We thank Reviewer#3 for this remark. As described in this revised version of the manuscript, the present study focused on the encapsulation of PZQ in PMMA-based microparticles and nanoparticles using suspension and miniemulsion polymerization strategies, respectively. Our findings showed that particle size distributions and composition can significantly affect the drug delivery profiles in the analyzed case, which can be important for developing efficient PZQ delivery systems. In particular, the obtained results showed that the PZQ encapsulation in pH-sensitive PMMA-co-DEAEMA particles can promote many competitive advantages for treatment of schistosomiasis, since this matrix provided better release profiles, reaching values close to 100 % of PZQ release in less than 3 h. This study also showed that microspheres based on DEAEMA provided the best absorption profile, displaying Cmax average value (1007 ± 83 ng/mL) that was 2.2 times higher than obtained for PZQ in the free form (432 ± 98 ng/mL). Based on all these positive and encouraging results, the polymer matrix based on PMMA-co-DEAEMA was selected as the best matrix among the investigated ones for preparation of a new pharmaceutical product.
Furthermore, the present study confirmed the promising potential of using polymer microparticles in the pharmaceutical industry. As a matter of fact, polymer microparticles can be used to improve the bioavailability of drugs and provide means to adjust the respective administration forms, constituting an important area of research for development of new and improved drug formulations. Moreover, microparticles can be utilized to turn medicine administration easier, especially for children. In the specific case of praziquantel, which has a bitter taste and must be administered in relatively high dosages, microparticles can be formulated into liquid or semi-solid forms to mask the drug taste, enhance the drug efficiency and make drug administration easier. This is particularly important for pediatric patients, who may face difficulties to swallow large pills and may demand fractionation of dosages. With help of microparticles, drug delivery can be customized to meet the specific needs of patients, thereby improving treatment outcomes and patient compliance.
